# Exploring Indigenous Community Conceptions of Parent Wellbeing: A Qualitative Analysis

**DOI:** 10.3390/ijerph20043585

**Published:** 2023-02-17

**Authors:** Arvind Ponnapalli, Tarita Fisher, Karen M. T. Turner

**Affiliations:** 1Darling Downs Health, Queensland Health, Toowoomba, QLD 4350, Australia; 2School of Psychology, The University of Queensland, St. Lucia, QLD 4072, Australia

**Keywords:** Aboriginal and Torres Strait Islander, Indigenous, social and emotional wellbeing, parent wellbeing, parenting

## Abstract

Using non-Indigenous perspectives of parental social and emotional wellbeing in the design and application of parent support programs can undermine program effectiveness as it may not account for Indigenous family structures and community values. With a clearer understanding of Indigenous parent wellbeing and its determinants, parenting interventions can be more appropriately designed and tailored to provide support for Indigenous families. This study utilised a community-based participatory action research approach involving collaboration between the research team, participants, and community advisory groups to explore Indigenous parents’ and carers’ conceptions of wellbeing. Participants’ cultural perspectives on parent wellbeing were collected through semi-structured focus groups and in-depth interviews (*N* = 20). Thematic analysis was undertaken using theory-driven and interpretative phenomenological analysis. Eleven themes emerged as risk and protective factors across three domains: child domain (i.e., school attendance and education, respect, routine, development), parent domain (i.e., role modelling, self-regulation of body, self-regulation of mind and emotions, parenting strategies), and context domain (i.e., connections to family and kinship, community, access to services). It is noteworthy that parents reported three super-ordinate intersecting themes across all domains: connection to culture, Country, and spirituality. In addition, Indigenous parents’ and carers’ conception of their own wellbeing is closely linked to their children’s wellbeing, their lived community context, and expected personal indicators. In recognising and working with this holistic view of Indigenous parent wellbeing, parent support programs can be optimally designed and implemented in Indigenous communities.

## 1. Introduction

The influences of parent wellbeing on the parent–child dyad [1,2,3] and early child development [4,5,6] have been explored extensively in the general population. The most commonly researched parent wellbeing indicators include parental depression [7], trauma [8], problem gambling [9], and parental stress [10]. Compared to the general population, Aboriginal and Torres Strait Islander peoples (henceforth respectfully and collectively referred to as Indigenous) and the Indigenous peoples of other high-income countries such as Canada, New Zealand, and the US continue to experience disproportionately poor physical health, mental health, and personal wellbeing [11,12]. The aetiology of these disparities, particularly physical morbidity and disability, are attributed to the social determinants of health over time due to a shared experience of colonialism [13,14].

In developmental psychology and the design of parenting programs, much of the early applied research was embedded in the epistemology of a Western tradition. Across geographical and socio-cultural norms, children’s developmental needs are predominantly universal [15]. Yet, different community settings pose unique sets of caregiving opportunities and adversities [16]. Furthermore, cross-cultural literature indicates considerable differences in the concepts of wellbeing, caregiving practices, and the concept of family between Indigenous cultures worldwide and Western cultures [17]. Nevertheless, the inferences from such a Western discourse continue to be uncritically generalised to Indigenous population [18,19,20].

There is lack of focus in research and practice towards understanding the components of Indigenous parents’ wellbeing from the perspective of early and preventative interventions. We propose that to promote Indigenous parental wellbeing (IPW) through parenting programs, it is vital first to understand the phenomenon of IPW and its components as perceived by Indigenous parents and caregivers.

Traditionally, the conceptualisation of parental mental health focused primarily on illness or disorders [21]. Consequently, evaluation studies of parent interventions relied on deficit-focused measures to evaluate parent wellbeing [22]. An expanding body of qualitative research on Indigenous wellbeing argues that the conceptualisation of wellbeing extends beyond this deficit-focus and should include positive wellbeing [23]. However, little is known about the contemporary conceptualisation of parent wellbeing, how its constructs might influence parents’ and child-specific outcomes, and if these constructs are sensitive to change in response to parent interventions [1].

Various evidence-based parenting support (EBPS) programs have demonstrated benefits for child physical, emotional, and behavioural adjustment by addressing modifiable parent-specific risk and protective factors, such as parenting knowledge, confidence, and skills [24,25,26]. With parenting and child outcomes as the primary focus, many parenting programs have shown secondary improvements in parent wellbeing indicators, such as parents’ depression, anxiety, and stress [22]. Previous research literature argues that interventions for children’s emotional, developmental, and behavioural concerns to include attention to parental wellbeing as well as parenting [27,28,29]. This may provide parents with a framework to assist their reflective functioning [30,31] and coping skills [29].

There is a clear imperative to address wellbeing in Indigenous communities. It is understood that the phenomenon of subjective social and emotional wellbeing (SEWB) in Indigenous Australians is not a single construct but is composed of several separable yet related factors [32,33,34,35]. However, the application of this holistic understanding continues to be inadequate in clinical practice. For example, due to the lack of well-validated instruments to assess SEWB, the measurement of wellbeing often depends on the basic psychometric properties of available instruments normed on the general population [36], which may have limited suitability [37]. This lack of relevant theory and outcome measures is one of the critical factors limiting the development and evaluation of interventions for Indigenous populations [36]. It is, therefore, difficult to articulate evidence-based practice in SEWB interventions, suggesting a need for an over-arching framework from which to approach low levels of SEWB, drawing on the concepts of ‘grief and loss’ and ‘healing’, and how high levels of social disadvantage have an impact on service utilisation and outcomes [38]. With a better understanding of IPW and its determinants, outcome measures and parent support initiatives can be appropriately designed and tailored for circumstances unique to Indigenous families and communities.

The aim of the current study was to explore the phenomenon of Indigenous parent wellbeing from parents’ and carers’ own perspectives, including key wellbeing domains and how they interact with each other (RQ1), and factors that promote or limit parent wellbeing (RQ2). We acknowledge the Indigenous belief that people’s stories are part of themselves and appreciate the participants’ trust in sharing their stories and personal experiences of wellbeing within their caregiving roles. The focus of this study was to extend current models of Indigenous SEWB to conceptualise parent wellbeing from the perspective of Indigenous families.

## 2. Methods

### 2.1. Research Design

This study adopted a qualitative research design, through focus-group discussions (FGDs) and in-depth interviews, to explore Indigenous parents’ subjective wellbeing experiences. The study utilised a multiphase design using an adaptive collective consensual data analytic procedure [39] in collaboration with the project’s community-based advisory group members to verify the IPW model and its domains and sub-themes (Figure 1).

### 2.2. Research Setting

This research was conducted in a discrete Indigenous community in South-East Queensland, Australia. The community was founded as a mission settlement in 1900 for Aboriginal and Torres Strait Islander peoples removed from all parts of the state. The population is less than 1500, and household income is half that of the rest of the Australian population [40]. As in most Indigenous communities, the population is young, with 36.9% of the community under 14 years of age. There is a disproportionately low percentage of adults in the community aged between the ages of 20 and 55, which is the typical age range of parenting [40]. The shortage of parent population is assumed to contribute to the high levels of disadvantage, child wellbeing issues, and overwhelming parenting burden on the older population in the community.

### 2.3. Research Team

The lead researcher (AP) is a male clinical psychologist of Indian heritage who has had ten continuous years of practice in the public health system in the Indigenous community where the current research was conducted. The co-researcher (TF) is an Aboriginal woman from the community who returned to the community 23 years ago to work in public health. She has over 20 years of experience in management and research in the health system. To ensure a culturally responsive evaluation [41], TF manually reviewed de-identified transcripts for inter-rater coding and validated the codebook for this study and summary themes. The co-researcher (KT) is a clinical psychologist with experience tailoring evidence-based parenting support programs to meet the lived context of Indigenous families in diverse communities. Throughout the study, researchers actively consulted with community-based advisory groups (made up of Elders, community-controlled health service, and community council members) to manage the researchers’ subjectivity and interference.

### 2.4. Participants and Recruitment Process

The eligibility criteria for participation included: (a) Aboriginal or Torres Strait Islander parent or carer; (b) at least one pre-teen child living in the home; (c) willing to participate in a small focus-group discussion (FGD) or semi-structured in-depth interview (IDI); and (d) resident of the community or neighbouring towns. The participant recruitment and data collection occurred between January and April 2020 before the COVID-19 lockdown. Recruitment strategies included incidental invitations to parents and carers at the local public library playgroup, community health service, and through word of mouth in the community. Study participants identified with seven different language groups, with diverse Country of origin. Table 1 outlines the participant demographic characteristics.

### 2.5. Procedure

#### 2.5.1. Data Collection

The participants’ parenting experiences within an Indigenous community and their subjective perspectives on parent wellbeing were collected through five FGDs (*n* = 13) and individual IDIs (*n* = 7). FGDs were offered, in keeping with a yarning methodology; however, community members were most comfortable with these being small groups consisting of family/kinship groups or work colleagues. Participants were comfortable to share their ideas in this setting. A brief demographic survey was employed to collect information about the sample and the representativeness of the parent population. An iPad audio recorder was used to record the conversations with the participants’ consent. Participants attended FGDs and IDIs in venues within the community which were convenient for them, and the lead researcher ensured privacy and confidentiality. As part of the information provided (participation information sheet) and consenting (consent form), participants were informed that if they should become upset or distressed during the process, they were free to pause or take a break or, if they preferred, they were free to withdraw without any explanation or disadvantage. Participants were also informed of the intended use of their stories and that they were entitled to receive feedback should they have chosen. The lead researcher and a research assistant used semi-structured questioning (Appendix A) within a yarning (consulting) methodology combined with Dadirri (deep listening) and Ganma (two-way knowledge sharing) [42].

#### 2.5.2. Data Analysis

This project utilised dualistic techniques to establish coding and thematic analysis through an interpretative phenomenological methodology [43]. Coding and thematic analysis involved both a theoretical or deductive approach [44,45] and an inductive approach [46]. Audio recordings were transcribed and verified by the lead researcher (AP). Although the formulation of preliminary codes and themes was theory-driven (i.e., based on the model of SEWB for Indigenous people [47]), emergent codes and themes were then identified through the IPA approach.

The research team adopted the steps illustrated by Roberts and colleagues [48]: (a) initial code sources developed through a literature review identifying a theoretical framework; (b) initial code development by application of raw qualitative data through a priori coding and IPA coding, code testing by co-author (TF), who was blinded to participant identity, and achieving code/theme saturation; (c) codebook development involving labelling the codes and grouping codes to recurring themes and domains; (d) codebook application with review by community-based advisory group members to confirm coding categories; and (e) conceptualisation of the phenomena using the theoretical framework and assumptions outlined earlier to develop a model of IPW.

The research team ensured the data analysis and interpretations were contextual and culturally responsive through data display to the community-based advisory group members, who verified the domains and sub-themes of the IPW model.

## 3. Results

Thematic saturation was determined by several considerations outlined by Braun and Clarke [49]. These included the ontological (i.e., “What is the nature of IPW?”) and epistemological (i.e., “What do Indigenous community members know about IPW?”) nature of the research questions, the conception of the codebook process, and consultations with the community-based advisory groups. This study’s thematic saturation was not based on a software-driven consensus between the coders. It arose from the depth of engagement with the data, consultations with the community-based-advisory group members, and a reflexive interpretation of themes from an ethnographic perspective.

Given the holistic concept of Indigenous wellbeing and interactionist assumptions [50], the thematic analysis was categorised into three super-ordinate domains: the child domain, the parent domain, and the context domain. Within each super-ordinate domain, several themes emerged. Participants also discussed three overlapping and intersecting themes, which parents believed were essential for all three domains. These included connections to culture, Country, and spirituality (see Figure 2 for the IPW model’s domains and themes). These are intrinsically interwoven with enablers and barriers (relating to child, parent, and context attributes) identified for each theme, as discussed below.

### 3.1. The Child Domain

A common view among participants was that their wellbeing is strongly linked to their children’s wellbeing: *“If my children are OK, then I’m really OK in myself. If my children are not OK, then I am not OK” (IDI_P5, mother)*. Four broad themes emerged from the interpretative phenomenological analysis of participants’ language and perspectives on child wellbeing: children’s school attendance and education, respect, routine, and development.

#### 3.1.1. School Attendance and Education

Several participants discussed that their own wellbeing is linked to their children attending school regularly and their academic achievement. One parent summarised:


*“[Children] go all the way through school. Make something of themselves in school, get a good education, get a good job and have a good life. Just education and being more respectful.”*
(FGD_P1 (mother))

##### Enablers

Participants described child-specific enablers for school attendance and education. For example, *“They [children] were excited for school… [name], he didn’t go to bed till late because he was that excited about going to kindy”* (FGD1_P1, mother). A father expressed a parent-specific enabler from his observations of working at a local school, that parents who look after their health, limit their substance use and gambling tend to ensure their children *“are at school and they pretty healthy”* (FGD2_P1, father). An Elder of the community expressed a context-specific enabler of this theme:


*“What I believe is the schools gotta give the parents a sense of ownership, a sense of belonging to the school, making the school friendly… they have two Aboriginal staff in the front desk. Now that’s a big tick in the box, because parents walk in and they see a friendly face. Previously, they saw all non-Indigenous people in the front desk and they gonna walk out feeling shame, feeling low self-esteem.”*
(IDI_P6 (grandfather))

##### Barriers

Regarding academic achievement, some parents, particularly those working full-time, expressed challenges in balancing commitment to their work and their children’s academic achievement: *“I don’t like when they get you know [low] marks at school sometimes, especially they just wanna learn because sometimes I feel like I am failing them”* (FGD2_P2, mother). Parents also expressed worry and challenges in buffering their children from context-specific risk factors, including exposure to youth suicides in the community and suicidality among their peers at school: *“They see and they hear what they done [youth suicides], so they act it out or they talk about it. You still hear a lot of it in the school”* (FGD1_P2, mother). A bereaved mother of three young children, who lost her oldest son to suicide, expressed a child-specific concern:


*“Yes, after September [son’s suicide] my children are struggling at school… [name] he actually run amok in the school. He was getting suspended every day and on re-entry, he would get suspended the same day.”*
(FGD4_P1 (grandmother))

#### 3.1.2. Respect

The term ‘respect’ was used interchangeably with Indigenous ‘culture’. Participants discussed the importance of instilling respect in their children through participation in their community events, discussing family values, teaching cultural heritage, and building a connection to Country and spirituality. Commenting on ‘respect’, participants reported experiences of positive wellbeing when their children demonstrated respect for themselves, their families, Elders, Country, and spirituality.

##### Enablers

As a parent-specific enabler, participants described teaching children respect as their parents and caregivers taught them. This parental responsibility ensures the transmission of cultural identity to the next generation: *“It [respect and culture] was taught to you by our parents, and it’s our part to pass it on to our children”* (FGD3_P1, grandmother). One participant explained the importance of teaching children their cultural identity:


*“Education was key for me, but my grandparents said also knowing who you are and where you come from, so knowing your mob, you know. Understanding where you fitting in this community, whether you Wakka Wakka or Gubbi Gubbi or your family got sent here from north Queensland or just knowing who you are is a big identity for wellbeing as well.”*
(IDI_P6 (grandfather))

##### Barriers

Several participants expressed their concerns of an intergenerational disconnection between their children and previous generations regarding the demonstration of respect, particularly for Elders. The comments below from a mother and from a grandmother encapsulate this generational gap that most participants echoed.


*“I was reared up with respect for Elders. Not now, these young ones they have lost respect. That’s the main thing I can talk about in this mission right now. Everybody lost respect for their Elders.”*
(IDI_P4 (mother))


*“Now, let’s think about where I live now is a special piece of ground because our grandparents lived there first and our mum and dad and our generation, my children, grandchildren, great grandchildren. Seven generations that share this piece of ground… So, kind of special…but I can’t see anyone. Any one of my children that will…share the same connection to this special piece of ground.”*
(FGD3_P3 (grandmother))

#### 3.1.3. Routine

Participants’ subjective experience of wellbeing appeared to be high when their children were in a ‘routine’. Participants, particularly single parents, discussed feeling ’relaxed‘ in their parenting role when their children were in daily routines.

##### Enablers

Participants’ descriptive language of their children in routine (child-specific enablers) included children ‘smiling’, ‘feeling happy’, and ‘in routine’ in the context of regular sleep time, coming home time from play, and regular school attendance. One mother expressed her delight: *“It’s stories they come home with too, eh, bring a smile to our faces too, things they do and stuff”* (FGD2_P2, mother). One grandfather discussed regular self-care activities as well as having family routines as a parent-specific enabler:


*“I was trying to take care of myself by getting in a routine of walking or exercising… basically, getting in a routine [both for parents and children], in a positive routine as well, is helpful.”*
(IDI_P6 (grandfather))

##### Barriers

Parents discussed context-specific challenges associated with disruptions to children’s and parents’ routine:


*“The violence happing around you, the alcohol, the drugs, everything just everywhere, not just in the community, it’s everywhere. Where I live…there is always fights there and I am trying to keep the children in the yard, but they see other children running out. It affects me because I am trying to keep them in the yard because I wanna keep them from them things.”*
(FGD1_P1 (mother))

In the context of recurrent ‘sorry business’ in the community, due to suicides and natural deaths, many participants described parent-specific challenges including prolonged grief, chronic fatigue, exacerbation of existing physical health conditions, and symptoms of depression impacting parents’ wellbeing and their ability to maintain their own and children’s routines.


*“We are starting to get back [to routine] as we used to before [son’s death to suicide]. We are not full on back there yet. We still have our peaks and downs…beforehand, kids have cereal or cook up a feed like bacon and eggs or whatever. But now, it’s just…they [kids] just help themselves.”*
(FGD4_P1 (grandmother))

#### 3.1.4. Development

Participants spoke of feeling ‘joy’ or feeling ‘happy’ in the context of ‘watching’ their children grow and develop. Participants interchangeably used the terms ‘growth’ and ‘development’ with inferences to different aspects including physical, emotional, social, and cultural development.

##### Enablers

Participants discussed several parent-specific enablers of child development, including parents spending time with their children, checking in on each aspect of children’s development and wellbeing, and guiding their children across their developmental stages to achieve maturity, both in their physical growth and developmentally appropriate skills acquisition. This enhanced their own sense of wellbeing:


*“What gives me the most joy is seeing the children grow and develop, seeing my grandchildren grow and develop, just checking in on their wellbeing you know and staying positive.”*
(IDI_P6 (grandfather))

The role of nurturing children’s development had major positive effects in some cases:


*“To be honest, I’ve been in jail all my life, I’ve been out now for the last 7 years and I’m happy to be home with my children, that makes me happy. It’s about time, I stay home, I share, yeah looked after my children and my partner and watch them grow.”*
(IDI_P5 (father))

##### Barriers

In the context of high rates of developmental and psychological vulnerability in local children, participants discussed vigilance for developmental concerns, and lived experiences of having limited access to service providers locally.


*“I worry about development down the line because their mother is too drunk. Drank while she was pregnant. So, it is still yet to determine whether learning abilities when she gets older and when she is ready for school.”*
(FGD3_P1 (grandmother))

### 3.2. The Parent Domain

A second super-ordinate domain of IPW emerged, the parent domain. Four themes emerged from the analysis of parents’ perspective of their personal wellbeing indicators: role modelling, self-regulation of their physical health (body), self-regulation of their mind and emotions, and parenting strategies.

#### 3.2.1. Role Modelling

Participants were aware that their children emulated parents and caregivers in all aspects of physical, emotional, social, and cultural wellbeing. An important indicator of parental wellbeing was their own role modelling of self-care to their children.

##### Enablers

In view of the high prevalence of chronic diseases among Indigenous families, one father explained how he ensured his children’s physical wellbeing by teaching them about healthy nutrition and exercise and by means of role modelling himself:


*“Especially in our communities, Indigenous communities that’s where physical activities is a must. Diet, nutrition is very important. What to eat, what not to eat, that’s teaching our kids at a young age, so they not caught up. You know lead them by example.”*
(FGD2_P1 (father))

A single mother with a stable part-time job described her role modelling to her children regarding social-emotional wellbeing and daily routine, going to school and work:


*“I hope they learn from me… what they see is what they gonna be doing and that’s why I hope I’m setting a good example, sitting home with them, see me going to work, when I say ‘come on now get up and go to school, coz I gotta go to work’.”*
(IDI_P5 (mother))

##### Barriers

Participants discussed the challenges of culturally incongruent examples of role modelling from themselves and others to their children. For example, *“Sometimes I get swearing and I don’t want to, coz when I swear, I don’t want the children to say what I’m saying. I don’t want him swearing ya know”* (FGD1_P3, mother).

#### 3.2.2. Self-Regulation: Body

One mother commented on her parenting values, including a desire to *“live longer for them [children], enjoy more play with my children”* (FGD1_P1). One father discussed the familial risks of physical health: *“Issues passed down from our families… diabetes is rife in Indigenous communities”* (FGD2_P1). Participants’ parenting values, combined with their own perception of risk, motivated them to self-regulate their physical wellbeing. One mother summarised the importance of ensuring personal physical wellbeing in parenting: *“If you don’t take care of your own health, you can’t be there to support your children the way you want to”* (FGD5_P1).

##### Enablers

Participants used descriptions such as ‘getting back to training’ (exercise), and ‘making healthy choices‘ (e.g., healthy diet, not smoking) as enablers to self-regulate their physical health. One participant discussed the importance of having access to health services (a context-specific enabler): *“Health [services] is needed. Community health service, a hospital, especially in our surroundings because of the issues [chronic diseases]”* (FGD2_P1, father).

One mother recalled her experiences of domestic violence, grief, and trauma. Although she had little understanding of her own body or emotional wellbeing, she intuitively held on to her spirituality and faith to survive.


*“[I] had fourteen babies and buried three sons from pregnancies to premature labour and lost my fourth pregnancy to a miscarriage. And even going through that, I had no understanding. It was pretty chaotic, simply because I had no understanding of what my body was going through physically or what my mental and psychological evaluation was doing to me. Just holding on spiritually to faith and in saying that it was my spirit that kept [me] alive.”*
(IDI_P1 (mother))

##### Barriers

One father reflected on causes of his heart attack at 29 years of age and also discussed his peers having heart attacks at a very young age due to unhealthy eating, excessive smoking, alcohol consumption, and illicit substance use:


*“I had heart attack from a lifestyle of unhelpful eating. That was at 29 years of age. There was another young bloke before me same age as me. Then after that I started having training and there was another bloke. He was only 30. I had to go to [a tertiary hospital]. I was there for three days. It was lifestyle related; diet related. After that my lifestyle changed.”*
(IDI_P7)

#### 3.2.3. Self-Regulation: Mind and Emotions

This theme discusses participants’ resilience to cope with stressors and problems, and their ability to manage thoughts and emotions, using positive coping strategies that protect and promote wellbeing. Parents discussed the following enablers and barriers to self-regulating their own mental and emotional wellbeing.

##### Enablers

Participants discussed both parent-specific and context-specific enablers to self-regulating their mental and emotional distress; for example, taking a consistent perspective with their culture and spirituality, engaging in a traditional (shared) bereavement during ‘sorry business’, family and kinship connections:


*“Just the children being there, because if they weren’t there we wouldn’t be as active…proactive. I think they keep us on our toes, they keep our minds in perspective, keep the whole family in perspective because they are our future and we have to guide them till they grow up to be the future for their children and so on.”*
(FGD2_P1, P2 (father, mother))


*“You can see a bit of if stress will arise…to come about, I’ll talk to my wife…or she will talk to me, we calm each other down, say ‘calm down’ and when we calm down we will talk about it [stress, problems] ask for solutions and because of our faith we pray then and that helps us through and we are fine then.”*
(FGD2_P1 (father))

##### Barriers

One father discussed his barriers to wellbeing: *“There is no support for fathers. Fathers turn to violence, and they go to jail”* (IDI_P7). References to the risk factors associated with mental and emotional issues outweighed other themes. A considerable portion of the discussion about parents’ mental and emotional challenges was driven by perpetual grief and loss, community violence, and substance use.


*“There are days when I start thinking about my son [who completed suicide] and everything. I actually lock myself away. That’s when children know something is wrong and know that I am having one of my days. They [younger children] come in the room and lay around. So, we just sit down and have a yarn and reminisce about the same.”*
(FGD4_P1 (grandmother))


*“I thought he would come around and talk [cousin who completed suicide]. I was sad for a point. I just used drugs to get over it [grief].”*
(IDI_P7 (father))

#### 3.2.4. Parenting Strategies

Parents described a range of parenting strategies inherited from generations before that were culturally endorsed within an Indigenous community context, and that were linked to their own sense of wellbeing. These strategies not only promoted child development, but also ensured transmission of culture and respect, and connections to Country and spirituality. Participants described a collective approach to parenting which involved shared responsibility and caring. This connection with family and kinship included transmitting cultural heritage and identity through Dreamtime stories and stories of previous generations and from their own lives: *“We tell them [Dreamtime stories]. They say, ‘Mum tell us what Nan tell you’*… *Those stories of how old people raised their children”* (FGD1_P1, grandmother). Participants also spoke of taking children on Country for camping, fishing, rodeo riding, and bush tucker (food and traditional medicines derived from native plants). Parents also promoted spirituality and talked about their faith, and encouraged faith practices such as family prayers, church attendance, and Sunday school, and talking about faith and respect.

There were many specific parenting strategies discussed, including teaching children by setting a good example (e.g., in physical activity and healthy eating), promoting independence with minimal support, being organised, talking to children, incidental teaching through role modelling, showing affection, giving instructions, redirection to prevent escalation traps, using logical consequences, dealing with misbehaviour immediately, setting rules, and establishing routines, particularly in the context of family stressors.

##### Enablers

Parents reported how parenting challenges were mitigated by the presence of supportive family and kin within the community:


*“When I’m like that [stressed] I need a break from my children, my dad, that’s when I said family will step in, my children I just send them down to my dad. I send my children to their grandparents.”*
(IDI_P4 (mother))

Participants described involving others in the nurturance and support of young children, and how older siblings and extended family had a central role, providing care and instruction.


*“Because we had that many children in the house, we all looked after one another and our parents had to go and work and it was a hard job, being the oldest I had to look after the youngest.”*
(FGD1_P2 (grandmother))

##### Barriers

Parents discussed high stress and challenges with consistency arising from caring for multiple children within their community context, which often resulted in parental fatigue. One mother reported moments of laxness arising from fatigue:


*“When we are feeling good, we have rules and they are clearly set when, can be firm when we are feeling good and strong. Other times just let them go to the lolly jar and let them stuff their faces and do whatever, until bedtime.”*
(FGD5_P1 (mother))

### 3.3. The Context Domain

The third super-ordinate domain that emerged was the ecological context within which families lived. This domain reflects participants’ environmental resources and the parent–child–context interactions that determined parents’ and children’s wellbeing. Three emergent subthemes were identified within this domain, including connections to family and kin, community, and services and programs.

#### 3.3.1. Family and Kin

The theme of connection to family and kin was the most coded theme in the data sets, indicating this as a strong factor of IPW. Parents predominantly sought help from immediate family, kin, or friends. This support-seeking was based on the intrinsic cultural values of family relationships and mutual trust:


*“Each family would have a very positive person at the head of the family… like my family for example, we might have a prominent person, a couple of prominent people in our family groups, we can go to them, but if you can’t, talk to your uncle or your aunt. Surely every family has someone they can relate to in their family, ya know. And they trust, ya know, someone who they trust, they can talk.”*
(IDI_P6 (grandfather))

##### Enablers

When parents’ subjective wellbeing was low, participants described how connections with family and kinship enabled them to improve their wellbeing: *“One of the family members will always show up…lighten up the day”* (FGD1_P1, mother). A grandfather and Elder of the community described his own community resources to maintain his wellbeing within his caregiving role:


*“I am a Wakka Wakka perso. I actually get a group of Elders I sorta consult or I look up to as role models and freely speak to them and they give me advice on how we should approach stuff. How we, how you feel, like a check in sorta thing. I try to do it in my role models or support networks.”*
(IDI_P6 (grandfather))

##### Barriers

In the context of perpetual ‘sorry business’ in the community, family and kinship systems were saturated with grief and loss. One mother shared:


*“I don’t know. I think everyone just doesn’t have any answers for each other [in grief]. And them seeing me just in my own world. I think they just don’t know how to [support]…yeah!”*
(FGD4_P2)

#### 3.3.2. Community

As a discrete, rural Indigenous community, participants commented on close relationships, and reliance on support from family (including community members as well as blood relations): *“This is a mission, we all family, we all here for each other”* (IDI_P4, mother).

##### Enablers

Many participants reflected on cultural and community support and resilience in the face of adversity: *“Every time I have my down moments, ya know well it’s a small community and one of my family members will always show up at my house”* (FGD1_P1, mother). A father with longstanding learning difficulties with numeracy and literacy continued to encourage his son by visiting the local library to encourage him to read books and have access to education. Another mother described after-school activities accessing the local Country:


*“After school we took them [children] to the tip yard, right on top of the hill there. They went up there, we buck rodeo riders and got our children into it and we go fishing… Take them for walks, tell them about gums, what we learned when we were young. My children go to church as well.”*
(FGD1_P1 (mother))

##### Barriers

Many participants referred to the negative impact of community issues including youth suicides, violence, drug and alcohol use, and the shortage of support services for parents and their children and youth. One mother, when asked about her dreams and hopes for her children, responded *“[To] get out of here”* (FGD4_P2, mother). As well as concern for the impact on children, participants directly linked such community issues to their own wellbeing.

#### 3.3.3. Access to Services

Many participants acknowledged the need for accessible services and programs in their community:


*“There’s a lot of issues involved, as you know in small communities… It happens everywhere, no one escapes it so I think it’s how we deal with it in our way of thinking… That’s where I see that it’s a real need for communities to have a service, a health service, educational facilities to address those issues because those issues are, like I said, they make a big impact on everyday life.”*
(FGD2_P1 (father))

##### Enablers

Participants reflected upon their previous experience of seeking help from local services. They identified desirable service provider attributes such as being non-judgemental, able to be trusted, building relationships, able to communicate well, and having a positive and non-deficit perspective: *“Talking, talking to someone you can trust. Also, to have a good listen, not to judge”* (IDI_P7, father). One mother described a service which has not only helped her to further her qualifications and employment opportunities but also provided parenting resources, which improved her parenting confidence and skills: *“They help with the studies and ask what you want to be… As a parent, it helps me, gives me confidence and skills, how to communicate, without negativity”* (FGD5_P1, mother).

##### Barriers

Several participants reflected on the attributes of community services, which contributed to poor sense of wellbeing. These included limited access to parenting support and other programs when needed, not having trusted accessible workers to yarn with, and a lack of fit-of-service delivery methods. For example,


*“I guess I could go [to services], like there are lots of services there, but it’s all over the phone and you don’t need that. You need to be sitting in front of someone and talk.”*
(IDI_P7 (father))

A common view that emerged was that Indigenous parents were hesitant to discuss their parenting due to the perception that cultural ways of parenting (e.g., community members and older children supervising and caring for young children) were deemed unacceptable by mainstream service providers and may culminate in child safety notifications or children being placed in out-of-home care. Such perceptions often contributed to limited uptake of parenting support in the community, even when support was available:


*“I think there is a lot of judging and people are afraid to be judged like that. You have a service coming to your home and you really need help, but you cannot ask for it, and they judge you and probably think, ‘Oh, are you capable of looking after your children?’”*
(FGD5_P1 (mother))

### 3.4. Summary of Findings

A summary of these findings can be found in Table 2, including IPW themes and their respective enablers and barriers.

## 4. Discussion

The goal of this study was to better understand Indigenous parent wellbeing (IPW) and its enablers and barriers from the perspectives of Aboriginal and Torres Strait Islander parents and caregivers within a discrete Indigenous community. The authors’ functional and personal reflexivity [51] in the process of data collection, analysis, and conceptualisation of the IPW model was managed by using a community-based participatory action research design [52,53], giving precedence to Indigenous methodologies [39], and adopting a partnership approach [54] by actively engaging with the community and cultural context throughout the process [55]. To our knowledge, this paper presents the first empirical study regarding the conceptualisation of the domains and themes of Indigenous parent wellbeing. The study is limited by its conduct in one discrete Indigenous community, potentially highlighting only some enablers and barriers to wellbeing. For example, the issue of racism was discussed but, in a primarily Indigenous community, was not expressed as a large factor in parent wellbeing. This may be different in other rural and urban settings, and the results should be interpreted with the community context in mind. The study’s strength is that it involved a cross-section of parents and carers, from teenage parents to great-grandparents, extended family, and kinship carers. The implications of the findings are discussed here in relation to future research and clinical practice, including the development and validation of an Indigenous parent wellbeing measure and tailoring of early intervention parenting programs to support parents and caregivers of young children in circumstances unique to Indigenous families and communities.

### 4.1. Conceptualisation of Indigenous Parent Wellbeing Model

The conceptualisation of the IPW model (RQ1) in this study (see Figure 2) was based on the interactionist [50] and holistic assumptions of Indigenous wellbeing [33,34]. Based on the findings, it was assumed that the subjective experience of Indigenous parents’ wellbeing within their role of caregiving was a result of a complex interaction of parent indicators (e.g., role modelling, physical, emotional and mental self-regulation, parenting strategies) with those of children (e.g., school attendance and education, respect, routine, development) and context indicators (e.g., family and kinship, community, and access to services). This interaction was further mediated by intrinsic regulatory factors (e.g., connections to culture, Country, and spirituality) [50]. Central to the concept of IPW was a reciprocal interaction of parents with their children and their ecological contexts in which the family system was embedded. It was assumed that when the parent–child–context congruence was poor [55] between any of these three domains, it was likely that parents and their children experienced a poor sense of subjective wellbeing.

This holistic approach to understanding parent wellbeing required looking beyond isolated parent-specific psychosocial indicators (e.g., depression, anxiety, stress, aggression, parenting styles) and allowed the consideration of culturally unique enablers and barriers of wellbeing (RQ2). Table 2 summarises specific attributes that served as enablers and barriers to IPW. The nature of parent–child interactions with these proximal and distal ecological contexts determined both parents’ and children’s wellbeing [56].

### 4.2. Implications for Research and Practice

By demonstrating rigour, validity, and reliability [48] in this qualitative study, the aim then is to translate Indigenous cultural knowledge into clinical practice.

#### 4.2.1. Development of an IPW Measure

One of the key factors limiting the development of evidence-based interventions for Indigenous populations has been a lack of well-validated instruments to assess SEWB and how it changes over time as a result of an intervention [36]. Future research in this series aims to develop and validate a brief, subjective measure of IPW to determine whether these domains can be empirically measured and whether such a measure may be a change-sensitive outcome measure for use in future research and clinical practice. We believe these identified domains of parent wellbeing lend themselves to a brief assessment measure and are exploring measure development and a delivery methodology that is sensitive to Indigenous ways of being and knowing (e.g., conversational and pictorial elements rather than a simple Likert scale).

#### 4.2.2. Tailoring of Parenting Support Programs

Rather than assuming that evidence-based programs developed and tested in diver-gent Western community contexts, or interventions developed locally in one Indigenous community, can be simply transplanted to diverse Indigenous community settings, the IPW model provides a structure for intake assessments, to aid development of a clinical formulation of each family’s context, strengths, and support needs. This will aid tailoring of any program to each family’s unique needs and goals, including a focus on wellbeing as well as parenting skills. Sensitivity to the intrinsic nature of connection to County, culture, and spirituality can only enhance culturally safe service delivery and build trust between practitioners and the families they work with. The IPW constructs can also form the target of public health campaigns promoting wellbeing and normalising the process of seeking parenting support, which could reduce the stigma often associated with accessing services. The authors are currently developing resources addressing the identified wellbeing domains including: universal intervention involving community radio public service announcements and social media content, and targeted preventive evidence-based parenting support interventions tailored to Indigenous parent wellbeing constructs. These will be trialled in the same community.

The key assumption from the current research findings is that in addition to personal wellbeing and parenting confidence, Indigenous parents’ subjective experience of wellbeing is related to their perceived ability to address their children’s wellbeing indicators (i.e., school attendance, education, respect, routine, and development) and their ability to access their contextual resources (i.e., family and kin, community resources, services, and programs). Future research should address these parent, child, and context indicators within tailored universal and targeted parenting interventions to ascertain if this translation of cultural knowledge to clinical practice has a measurable change in parent and child wellbeing.

## 5. Conclusions

Australian Indigenous communities are diverse in cultural expression, connection to Country, and language [57]. Any consideration of the generalisability of the IPW model from this study to other communities may require considerations of the emic and derived etic approaches to cultural research [58]. Applying John Berry’s [58] distinctions to the current research, this study aimed to represent the perspectives of Indigenous parents and caregivers through focus group discussions, interviews, and through community consultations with the advisory groups (i.e., emic approach). The current study subsequently adopted a type of analysis and interpretation to fit the perspectives of the participants and the meaning of phenomena (i.e., derived etic approach). Some scholars conclude that there should be no attempt to generalise qualitative study findings [59]. However, Berry proposed that a phenomenon (e.g., IPW) studied in one community may be applied to another community only when the phenomenon was shown to be functionally equivalent [60]. While the unique enablers and barriers of IPW may vary from community to community, the broader assumption of the child–parent–context coherence may be generalisable to other Indigenous communities.

The current study is the first in a series of studies that aim to test the hypothesis that addressing Indigenous parent wellbeing indicators within evidence-based parenting support may likely results in enhanced child and parent outcomes. The empirical evaluation and clinical utility of the IPW model will be tested through the development and validation of an Indigenous parent wellbeing measure and evaluation of a multilevel community-wide parent support intervention aimed at promoting parent wellbeing and positive parenting. As intervention studies point out the causal connection between parent wellbeing and early childhood outcomes [61], adopting a holistic view of parent wellbeing in intake, program design, and therapeutic relationship is an endeavour with potentially far-reaching benefits.

## Figures and Tables

**Figure 1 ijerph-20-03585-f001:**
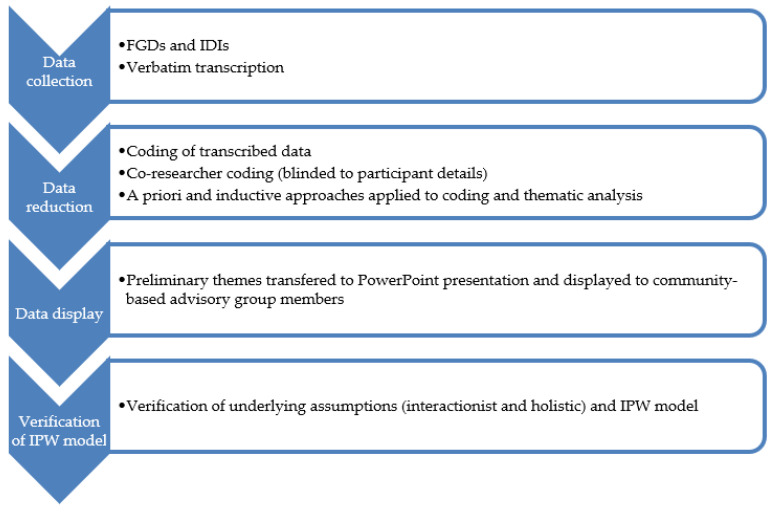
Collective consensual data analytic process.

**Figure 2 ijerph-20-03585-f002:**
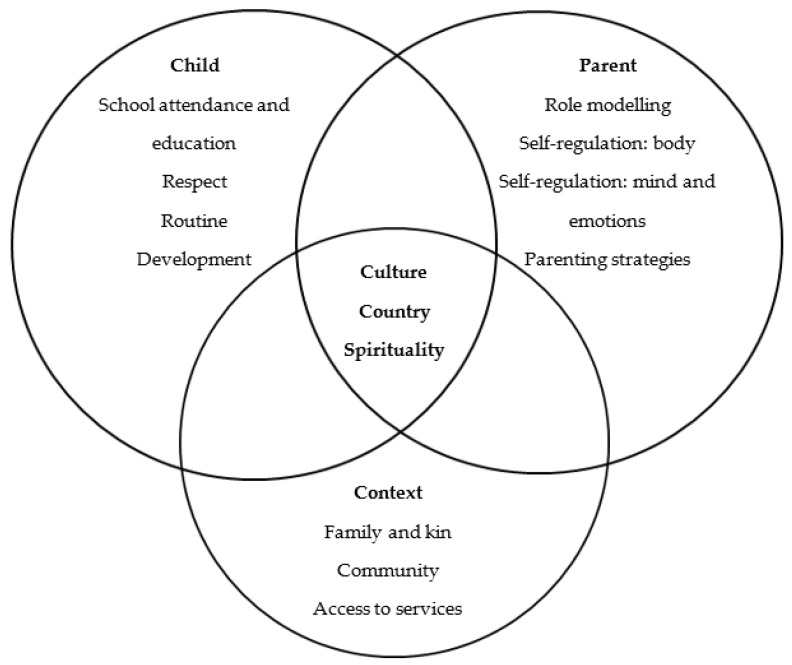
Indigenous Parent Wellbeing Model.

**Table 1 ijerph-20-03585-t001:** Participant demographic characteristics.

ID No.	Age	Sex	Parent Role	Primary Language Group	No. of Children under 12 Years
FGD1_P1	41	F	Mother, grandmother	Wakka Wakka	5
FGD1_P2	49	F	Kinship carer	Wakka Wakka	1 *
FGD1_P3	36	F	Mother	Kabbi-Kabbi, Kullalli	1 *
FGD2_P1	50	M	Father	Wakka Wakka	3
FGD2_P2	41	F	Mother	Wakka Wakka	3
FGD3_P1	63	F	Grandmother	Koa	2
FGD3_P2	76	F	Great-grandmother	Koa	1
FGD3_P3	66	F	Grandmother	Koa	1
FGD4_P1	39	F	Mother, grandmother	Wakka Wakka	3
FGD4_P2	18	F	Mother	Wakka Wakka	1
FGD5_P1	29	F	Mother	Kullilli, Komet	3
FGD5_P2	42	F	Aunt, kinship carer	Bindal	2
FGD5_P3	27	F	Mother	Wakka Wakka	3
INVP1	41	F	Mother, grandmother	Wakka Wakka	
IDI_P2	23	M	Father	Kabbi Kabbi	1 *
IDI_P3	25	F	Mother, kinship carer	Kabbi Kabbi, Wangan, Jagalingou	3
IDI_P4	38	F	Mother	Wakka Wakka	4
IDI_P5	46	M	Father	Wakka Wakka	3
IDI_P6	52	M	Grandfather	Wakka Wakka	2 *
IDI_P7	31	M	Father	Wakka Wakka	1

* Parent has intermittent care of the child/ren under 12 years of age.

**Table 2 ijerph-20-03585-t002:** Summary of findings.

Domain (RQ1)	Themes	Sub-Themes: Enablers (RQ2)	Subthemes: Barriers (RQ2)
Child	School attendance and education	Children’s engagement and participation in school Parental self-care Culturally responsive schools	Perpetual sorry business Negative peer influences at school Child developmental issues Parental work–life balance
	Respect	Parental responsibility to teach respect Connection to culture, Country, spirituality	Generational gap Children being peer-oriented vs. parent/culture
	Routine	Parental role modelling and self-regulation Healthy child development	Exposure to D&FV, drugs and alcohol use Perpetual sorry business Grief and loss
	Development	Normal growth and development. Access to health services and check-ups	Developmental delays. Material ill-health (e.g., smoking, antenatal alcohol use, subject to D&FV)
Parent	Role modelling	Parents setting an example Parents demonstrating self-care	Parents setting culturally incongruent examples (e.g., swearing) Parents engaging in substance misuse, D&FV, problem gambling
	Self-regulation: physical health (connection to body)	Parents engaging in health behaviours, e.g., exercise, healthy diet, smoking cessation Parenting values and goals, e.g., longevity, enjoy play with children, being a part of children’s future	Harmful smoking Chronic diseases Depression, anxiety, D&FV
	Self-regulation: mental health (connection to mind and emotions)	Traditional ways of bereavement. Parental perspective taking in view child development Engaging in faith practices, e.g., prayer, church attendance	Parental disengagement from family and kin Substance misuse Grief and loss Youth suicides in the community
	Parenting strategies	Shared parenting Transmission of culture—Dreamtime stories, storytelling. Talking to children. Spending time with children Incidental teaching Logical consequences	Grief and loss Parental fatigue Perceived judgment from service providers
Context	Family and kin	Connection with family and kin	Family and kin overwhelmed by perpetual sorry business
	Community	Connected community Access to community resources (e.g., community events, local radio, bush, creeks)	Neighbourhood violence, exposure to drugs and alcohol use Shortage of support services for parents and children
	Access to services	Access to culturally responsive hospital and health services	Shortage of local social emotional wellbeing services for parents and children

## Data Availability

The data that support the findings of this study are available on request from the corresponding author. The data are not publicly available due to privacy or ethical restrictions.

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
