# Peer review of "Exploring Indigenous Community Conceptions of Parent Wellbeing: A Qualitative Analysis"

_ijerph, 2023, doi:10.3390/ijerph20043585_

Round 1

Reviewer 1 Report

I very much enjoyed reviewing your well done article and have only two or three suggestions for you. First, since your findings are suggestive and indicate a well reasoned strategy to the concern you address, I suggest arguing in line 11 that you hope your results will result in MORE appropriately designed and tailored strategies. I would suggest you delete the word optimal in line 12 for the same reason. The current wording in each case does not appear to be consonant with with your aims or findings. Second, I was puzzled by why you used focus groups  since they were apparently quite small (line 146)? Why not just do interviews? What did you gain from your FG's? In any case, I think this issue deserves two or three sentences to provide readers clarification on your rationale, process and findings regarding FG's. Third, I wonder if "themes" is the correct word in line 77? Your subject is emotional well being and so perhaps "factors" or "characteristics" would serve better here? Fourth, where are the parents alluded to in line 115 and forward? Your Table 1 does not suggest that you could not locate folks? Fifth, What is a "bush tucker?" (Line 426). A long walk? Last, I suggest you delete "subjective" in Line 545. It is unneeded.

Author Response

Dear Reviewer 1,

Thank you so much for your positive feedback. We sincerely accept your detailed and insightful comments and suggestions. We have addressed the comments as follows.

Reviewer 1:  

First, I suggest arguing in line 11 that you hope your results will result in MORE appropriately designed and tailored strategies. I would suggest you delete the word optimal in line 12 for the same reason. The current wording in each case does not appear to be consonant with your aims or findings.

Authors’s response:

We appreciate this nuanced understanding and have updated the text accordingly.

Reviewer 1:

Second, I was puzzled by why you used focus groups  since they were apparently quite small (line 146)? Why not just do interviews? What did you gain from your FG's? In any case, I think this issue deserves two or three sentences to provide readers clarification on your rationale, process and findings regarding FG's.

Authors’s response:

We are happy to provide more detail here on the community experience and have added the following on lines 148-151: “FGDs were offered, in keeping with a yarning methodology, however community members were most comfortable with these being small groups consisting of family / kinship groups or work colleagues. Participants were comfortable to share their ideas in this setting.”

Reviewer 1:

Third, I wonder if "themes" is the correct word in line 77? Your subject is emotional well being and so perhaps "factors" or "characteristics" would serve better here?

Authors’ response:

 We have replaced ‘themes’ with ‘factors’ (please see line 75).

Reviewer 1:

Fourth, where are the parents alluded to in line 115 and forward? Your Table 1 does not suggest that you could not locate folks?

Authors’s response:     

This statement refers to the general population demographic of our research setting rather than the sample in this study. We have clarified this on line 112-113 (“There is a disproportionately low percentage of adults in the community aged between the ages of 20 and 55…”). It is true that we were able to recruit a sample from a diverse age range.

Reviewer 1:

Fifth, What is a “bush tucker?” (Line 426). A long walk?

Authors’ response:

We have clarified this term on line 431 – it refers to food and traditional medicines derived from native plants.

Reviewer 1:

Last, I suggest you delete "subjective" in Line 545. It is unneeded.

Authors’ response:

Thank you, the text has been edited (line 553).

Reviewer 2 Report

Thank you for giving me an opportunity to review enclosed manuscript. It deals with interesting, not well-known and described issue, which could give useful insights for policy makers.

Minorities' well-being is an important goal to taking into account programing social and economic policies. Thus, I'd like to underline all of the Authors efforts to indetify the research gap, using necessary references and publications.

Moreover, the concept of the survey and it's organisation is well-planned and reasonable. Intergenerational connections and opinions has given full perspective of the issue. Presentation of the results (including enablers and barriers) is compelling.

One specific point how to improve the value of the presented mauscript is to develop and expand Discussion and Conclusion sections. Basicly, please add Authors' perspective regarding to "whether these domains can be empirically measured" and please write more in the text about the future fields of Authors' research in this case (line 635-640). It could be intersting for future readers of the manuscript.

Author Response

Dear Reviewer 2,

Thank you for your encouraging comments and helpful feedback. We have responded to the comments as follows.

Reviewer 2:

One specific point how to improve the value of the presented manuscript is to develop and expand Discussion and Conclusion sections. Basically, please add Authors' perspective regarding to "whether these domains can be empirically measured"

Authors’ response:

We have added detail in lines 604-608: “We believe these identified domains of parent wellbeing lend themselves to a brief assessment measure and are exploring measure development and delivery methodology that is sensitive to Indigenous ways of being and knowing (e.g., conversational and pictorial elements rather than a simple Likert scale).”

Reviewer 2:

and please write more in the text about the future fields of Authors' research in this case (line 635-640). It could be interesting for future readers of the manuscript.

Authors’ response:

Thank you for your interest in our ongoing work. We have added more detail on lines 621-625: “The authors are currently developing resources addressing the identified wellbeing domains including: universal intervention involving community radio public service announcements and social media content; and targeted preventive evidence-based parenting support interventions being tailored to wellbeing constructs. These will be trialled in the same community.”

And further at line 653-656: “Empirical evaluation and clinical utility of the IPW model will be tested through the development and validation of an Indigenous parent wellbeing measure and evaluation of a multilevel community-wide parent support intervention aimed at promoting parent wellbeing and positive parenting.”

Reviewer 3 Report

I had the opportunity to review an interesting paper about indigenous community conceptions of parent wellbeing. The paper is very well written in my opinion, I only have two small comments for the authors.

Introduction:

- I am not sure if the division into subchapters (1.1. – 1.3.) is necessary.

Discussion:

- I am thinking if it might make sense to move Table 2 to the Results section.

Author Response

Dear Reviewer 3,

Thank you for your supportive comments. We have addressed your suggestions as follows.

Reviewer 3:

I only have two small comments for the authors. Introduction: I am not sure if the division into subchapters (1.1. – 1.3.) is necessary.

Authors’ response:

We have removed the subheadings in the introduction as suggested.

Reviewer 3:

Discussion: I am thinking if it might make sense to move Table 2 to the Results section.

Authors’ response:

Thank you for this suggestion, we have moved Table 2 to the end of the Results section and refer back to it in the discussion.
